# HyperCLIP: Prompt-Conditioned Image Encoders for Contrastive Vision-Language Pre-training

## Abstract

CLIP-style image encoders are trained to be discriminative for every category set a user might supply, since the category set is unknown at training time. This requirement is more demanding than the one any single deployment actually imposes, and is part of why small image encoders underperform large ones on zero-shot classification. In CLIP, the class prompts available at inference are used only to define the classifier head; we argue they carry more task structure than this role exposes, enough to also modulate the image encoder's feature extraction through a small channel (BatchNorm scale and bias). We provide evidence for this view by introducing **HyperCLIP**, a contrastive pre-training architecture in which a hypernetwork generates the BatchNorm scale and bias of a small image encoder directly from the class-prompt embeddings produced by the text encoder, with all three components trained jointly under the SigLIP loss. Across eight small vision backbones, HyperCLIP improves zero-shot accuracy over a matched SigLIP baseline by up to 3.3% on ImageNet-1K and 5.6% on CIFAR-100; the gains concentrate in BatchNorm-rich backbones, are equivalent to one step up the EfficientNet scaling ladder, and recover roughly half of what supervised BatchNorm fine-tuning can achieve, without any task labels and with no added inference-time cost. We add controls that scope how much of the gain reflects prompt content rather than the added training-time capacity.

## 1 Introduction

In a CLIP-style contrastive model (Radford et al., 2021; Zhai et al., 2023), the image encoder must produce embeddings that are discriminative for any category set a user might supply; the category set is not known until inference. This requirement is both strict and wasteful: at deployment, a user provides a specific set of class prompts, and the encoder needs to discriminate only that set. The mismatch is part of why small image encoders underperform large ones on zero-shot classification: a small encoder cannot solve the open-vocabulary problem at training time, even though it would be entirely capable of solving the closed-vocabulary problem it actually faces at inference. The standard responses to this gap operate post-hoc, by compressing a large encoder back down through pruning, quantization, or distillation (Sun et al., 2023; Dettmers et al., 2022; Frantar & Alistarh, 2023; Dai et al., 2022; Wu et al., 2023; Vasu et al., 2024); they accept the open-vocabulary framing and try to absorb its cost.

In CLIP's standard zero-shot pipeline, the class prompts enter only at the very end: they define the classifier through their dot product with the image embedding, and the image encoder is identical for every label set. We claim the prompts carry more task structure than this classifier-defining role exposes: enough to also modulate the image encoder's feature extraction, through a small and well-localized channel (BatchNorm scale and bias). If that is right, a small image encoder should be able to reach much closer to a large one by routing the prompt embeddings into its forward pass, not by being pruned or distilled but by being given a use for task information that CLIP currently sends only to the classification head. We introduce **HyperCLIP**, a contrastive pre-training architecture that operationalizes this claim. A hypernetwork takes the prompt embeddings produced by the text encoder and outputs the BatchNorm scale and bias of a small image encoder; all three networks are trained jointly under the standard SigLIP loss. At inference, the hypernetwork runs once on the user's class prompts and installs the resulting parameters into the image encoder, which is

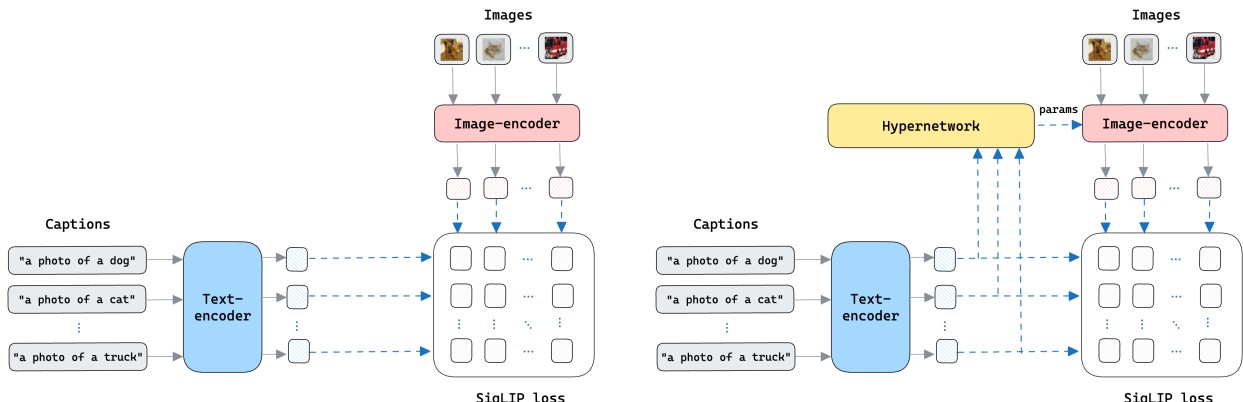

Figure 1: Architecture overview. (Left) The CLIP/SigLIP architecture: a text encoder $\mathcal{G}$ and image encoder $\mathcal{F}$ are trained against a contrastive objective. (Right) HyperCLIP: a hypernetwork $\mathcal{H}$ maps the text embeddings $Y$ produced by $\mathcal{G}$ to the normalization scale and bias parameters of $\mathcal{F}$. At inference, $\mathcal{H}$ is run once on the class prompts to produce a task-specialized image encoder.

then used as a zero-shot classifier in the usual way. The framework is intentionally narrow: we condition a small, well-understood control surface on a signal the model already produces, so that the marginal effect of conditioning can be measured against the added capacity directly. We do so with a capacity-matched control and prompt-perturbation tests (Appendix L); as those controls show, a nontrivial part of the gain reflects the added training-time capacity rather than prompt content, and we scope our interpretation accordingly.

Across eight small vision backbones trained on 128M samples of DataComp (Gadre et al., 2024), HyperCLIP improves zero-shot ImageNet-1K accuracy over a matched SigLIP baseline by up to 3.3% and CIFAR-100 accuracy by up to 5.6%; the gain is consistent across distribution-shift, retrieval, and fairness benchmarks, and survives linear probing. Three facts about *where* the gain comes from are informative. First, the gain concentrates in BatchNorm-rich backbones: LayerNorm-only and BN-poor backbones gain little, consistent with BatchNorm being the operative task channel. Second, the gain is roughly equivalent to one step up the EfficientNet scaling ladder (HyperCLIP-B0 matches SigLIP-B1, and HyperCLIP-B1 outperforms SigLIP-B2), which is the gain we would expect if the small encoder were being relieved of part of the open-vocabulary requirement, though Appendix L shows this scaling-equivalent gain is equally consistent with a capacity or regularization effect. Third, against a supervised upper bound that fine-tunes the baseline's BatchNorm parameters on each downstream task, HyperCLIP recovers roughly half of what is achievable from BN alone, without any task labels. We unpack these findings in Section 5.

## 2 Preliminaries

**Contrastive vision-language pre-training.** CLIP (Radford et al., 2021) learns an image encoder $\mathcal{F}\colon \mathbb{R}^{B \times I} \to \mathbb{R}^{B \times D}$ and a text encoder $\mathcal{G}\colon \mathbb{R}^{B \times C} \to \mathbb{R}^{B \times D}$ by maximizing the similarity of matched image-caption pairs and minimizing it for mismatched pairs in a batch of size $B$. The two encoders share an embedding dimension $D$, which lets a trained model be used as a zero-shot $K$-way classifier: given an image embedding $x \in \mathbb{R}^D$ and a matrix of class-prompt embeddings $Y \in \mathbb{R}^{K \times D}$, the prediction is $\arg\max_i (Yx)_i$. SigLIP (Zhai et al., 2023) replaces CLIP's softmax-based loss with a sigmoid loss that treats every pair independently:

$$\mathcal{L}_{\text{SigLIP}}(\mathbf{X}, \mathbf{Y}) = -\frac{1}{B} \sum_{i=1}^{B} \sum_{j=1}^{B} \log \sigma(Z_{ij}[\eta\langle \mathbf{x}_i, \mathbf{y}_j \rangle + \zeta]), \tag{1}$$

where $Z_{ij} \in \{+1, -1\}$ marks matched/mismatched pairs and $\eta, \zeta \in \mathbb{R}$ are learnable scalars. The sigmoid form makes the loss memory-efficient at large batch sizes and stable at small ones; we use it as the training objective for both HyperCLIP and our baselines. HyperCLIP is orthogonal to the specific choice of contrastive

objective; we use SigLIP because it is the most widely adopted CLIP variant in the small-batch regime (Appendix K).

**Hypernetworks.** A hypernetwork $\mathcal{H}(X_h; \Phi)$ produces the parameters $\Theta$ of a target "mainnet" $\mathcal{M}(X_m; \Theta)$ from a conditioning input $X_h$ (Schmidhuber, 1992; Ha et al., 2016; Denil et al., 2013; Bertinetto et al., 2016; Jia et al., 2016). Hypernetworks are known to be difficult to train because their outputs directly shape the loss landscape and there is no canonical initialization scheme (Chang et al., 2023; Beck et al., 2023; Ortiz et al., 2023). In HyperCLIP we sidestep these difficulties by restricting the hypernetwork to predict only the affine normalization parameters of the image encoder; we defer a comparison with the broader literature on hypernetworks and adjacent methods to Section 6.

## 3 HyperCLIP

### 3.1 Motivation

Consider the standard zero-shot use of CLIP. To classify an image into one of $K$ categories, the user writes a text prompt for each category, embeds the prompts with the CLIP text encoder, embeds the image with the CLIP image encoder, and predicts the nearest prompt in embedding space. For this to work, the image encoder must produce embeddings that are discriminative for *any* set of category prompts a user might supply, since the prompts are not known at training time. This requirement is the proximate reason CLIP image encoders are large.

But the image encoder does not need to be discriminative for every category set *simultaneously*: at inference, the user supplies a specific set of prompts, and the encoder only needs to be discriminative for that one set. If the encoder could be re-tuned at inference time to the specific category prompts it is about to see, a much smaller backbone could plausibly suffice. HyperCLIP implements this re-tuning by routing the prompt embeddings through a learned hypernetwork that produces a small set of image-encoder parameters.

### 3.2 Architecture

A HyperCLIP model has three components:

1. An **image encoder** $\mathcal{F}(\cdot; \Theta_{\text{fixed}} \cup \Theta')$ with the same functional form as a CLIP image encoder but substantially smaller. Its parameters are split into a fixed set $\Theta_{\text{fixed}}$ (convolution filters and MLP weights, shared across all inputs) and an adapted set $\Theta'$ (the normalization scale and bias parameters), which is produced by the hypernetwork.

2. A **text encoder** $\mathcal{G}$ with the standard CLIP causal-Transformer architecture, taken unchanged from Radford et al. (2021).

3. A **hypernetwork** $\mathcal{H} \colon \mathbb{R}^{B \times D} \to \mathbb{R}^{|\Theta'|}$ that maps a set of text embeddings to the adapted parameters $\Theta'$.

A forward pass on a batch of images $\mathbf{i}$ and captions $\mathbf{c}$ proceeds as:

$$\mathbf{y} = \mathcal{G}(\mathbf{c}), \qquad \boldsymbol{\theta}' = \mathcal{H}(\mathbf{y}), \qquad \mathbf{x} = \mathcal{F}(\mathbf{i};\ \Theta_{\text{fixed}} \cup \boldsymbol{\theta}'). \tag{2}$$

The image and text embeddings $\mathbf{x}, \mathbf{y}$ are then fed into the SigLIP loss in Equation 1. Gradients flow through $\mathcal{H}$ back to both itself and the text encoder, and through $\mathcal{F}$ back to $\Theta_{\text{fixed}}$ and (through $\mathcal{H}$) to the text encoder. All three networks are trained simultaneously.

### 3.3 Design choices

**What parameters to adapt.** In principle the hypernetwork could output every parameter of the image encoder; in practice, even small vision backbones contain millions of parameters and an output head of that

dimensionality is impractical and hard to train. We restrict $\Theta'$ to the *batch- or layer-normalization scale and bias parameters* of the image encoder. These are an attractive target for two reasons. First, they are small: the backbones we consider have on the order of $10^4$–$10^5$ such parameters (Table 1), keeping the hypernetwork output head tractable. Second, they are expressive: Frankle et al. (2020) showed that adjusting BatchNorm scale and bias alone is sufficient to obtain non-trivial accuracy in a deep network with otherwise frozen random weights, suggesting they provide a powerful control surface. We confirm empirically (Appendix G, Table 8) that hypernetwork prediction of convolutional or linear-layer weights did not beat the baseline; among normalization types, BatchNorm-based backbones benefit most, with LayerNorm-based ones (e.g., EdgeNext) gaining less.

**Hypernetwork architecture.**   The hypernetwork must accept a variable number of prompt embeddings and must be invariant to their order: the prompt list for a classification task has no canonical ordering, and the number of prompts varies between tasks. A Transformer with no positional encoding and no causal mask satisfies both requirements. We use a 12-layer non-causal Transformer (width 768, 8 heads, FFN dimension 2560, GELU, dropout 0.1) whose tokens are the prompt embeddings. We then average over tokens and pass the result through a bottleneck linear layer, a LayerNorm, and an output feedforward layer $FF_{\text{output}}$ whose output dimension equals $|\Theta'|$. This architecture is illustrated in Figure 2. The Transformer's depth is not essential to the effect: replacing it with a single linear map from the mean prompt embedding retains most of the gain (Section 5.5, Figure 4). We therefore treat the depth as an optional refinement rather than the load-bearing component, and do not attribute the method's benefit to the Transformer's expressivity.

**Stabilizing training.**   Hypernetworks are known to be difficult to train end-to-end (Chang et al., 2023; Beck et al., 2023). We found three design choices critical to making HyperCLIP train reliably. (i) Restricting the hypernetwork to predict only normalization scale and bias (small, well-understood affine parameters) greatly reduces the difficulty of the prediction problem. (ii) The Transformer-based hypernetwork inherits standard, well-tested initialization. (iii) We use PyTorch's functional API to inject predicted weights into the image encoder during each forward pass, which avoids the bookkeeping that ad-hoc weight replacement requires. We also constrain the scale parameters $\gamma$ to be positive via an exponential reparameterization and use running estimates of the batch statistics. An optional learnable weight-scale parameter $S_w$ applied to the hypernetwork output gave small additional gains (Appendix E).

**Image encoders.**   HyperCLIP is agnostic to the choice of image encoder. We evaluate eight small vision architectures: EfficientNet-B0/B1/B2 (Tan & Le, 2019), MobileNetV3 M0/M1 (Howard et al., 2019), TinyNet T0 (Han et al., 2020), EdgeNext E0 (Maaz et al., 2022), and MobileViT V0 (Mehta & Rastegari, 2021). Table 1 summarizes each backbone's parameter count, input resolution, type and number of adapted parameters, hypernetwork bottleneck dimension, and the relative change in training throughput induced by adding the hypernetwork.

Figure 2: The HyperCLIP hypernetwork. Prompt embeddings from the text encoder are processed by a noncausal Transformer, averaged, passed through a bottleneck, and projected to the normalization scale and bias parameters of the target image encoder.

### 3.4   Training and inference

**Training.**   For each image-caption mini-batch, we run the forward pass above and minimize $\mathcal{L}_{\text{SigLIP}}(\mathbf{X}, \mathbf{Y})$ end-to-end. The hypernetwork is trained jointly with both encoders. The parameters $\Theta_{\text{fixed}}$ of the image

Table 1: Image encoders evaluated in this paper. "# Adapt" is the number of normalization scale and bias parameters predicted by the hypernetwork; "Type Adapt" indicates which normalization layers are present (BN: BatchNorm, LN: LayerNorm, GN: GroupNorm); "Throughput $\Delta$" is the percentage change in training throughput from adding the hypernetwork.

| Backbone | B0 | B1 | B2 | M0 | M1 | T0 | E0 | V0 |
|---|---|---|---|---|---|---|---|---|
| Params (M) | 4.6 | 7.2 | 8.4 | 4.9 | 2.0 | 1.7 | 7.6 | 4.7 |
| Input size | 224 | 240 | 260 | 224 | 224 | 152 | 320 | 224 |
| # Adapt (K) | 42.1 | 62.1 | 67.6 | 24.4 | 12.1 | 17.1 | 8.8 | 15.5 |
| Type Adapt | BN | BN | BN | BN | BN | BN | LN | BN+GN |
| Bottleneck dim | 285 | 193 | 177 | 491 | 577 | 512 | 256 | 512 |
| Throughput $\Delta$ (%) | $-3.3$ | $-11.2$ | $-18.8$ | $+0.2$ | $+0.2$ | $-12.7$ | $-47.7$ | $+6.8$ |

encoder are updated by gradient descent in the usual way; the normalization parameters $\Theta'$ are not stored as trainable variables of $\mathcal{F}$ at all, but are regenerated by $\mathcal{H}$ each step. The text encoder is updated both directly through its contribution to the loss and indirectly through its contribution as input to $\mathcal{H}$.

**Inference.** Once trained, HyperCLIP is used as a zero-shot classifier in the same way as CLIP, with one extra step. Given a list of $K$ class prompts:

1. Embed the prompts with the text encoder to get $Y \in \mathbb{R}^{K \times D}$.

2. Run the hypernetwork *once* on $Y$ to obtain $\Theta' = \mathcal{H}(Y)$ and install these parameters into the image encoder.

3. Discard the text encoder and hypernetwork. Classify each new image $\mathbf{i}$ as $\arg\max_i (Y\mathcal{F}(\mathbf{i}; \Theta_{\text{fixed}} \cup \Theta'))_i$.

Because the text encoder and hypernetwork are used only once per task, their cost is amortized over every image classified afterwards. The per-image inference cost is exactly that of the small image encoder $\mathcal{F}$: HyperCLIP adds no parameters or operations to the deployed classifier relative to a SigLIP model with the same backbone. The only per-task cost is a single forward pass of the text encoder and hypernetwork over the $K$ prompts, plus storage of the resulting normalization parameters ($|\Theta'|$ scalars, on the order of $10^4$ to $10^5$; Table 1). If the label set changes, only this one-time step is repeated and the per-image cost is unchanged. HyperCLIP is therefore best suited to deployments where a label set is reused across many images, and least advantageous where it changes on every image.

For settings with task-specific training data, we additionally fine-tune a linear classification head on top of $\mathcal{F}$'s features (i.e., linear probing), with the head initialized to the prompt embeddings $Y$ and trained against the cross-entropy loss.

## 4 Experiments

We evaluate HyperCLIP along four axes: zero-shot classification across a broad benchmark suite, robustness under distribution shift, fairness across subgroups, and behavior after task-specific fine-tuning. We also report ablations over the hypernetwork architecture, batch size, training-data scale, and a comparison with pruning. Our goal in every comparison is to isolate the marginal effect of the hypernetwork: we train each HyperCLIP model and a matched SigLIP baseline on the same data, with the same image and text encoder architectures, batch size, and number of optimizer steps.

### 4.1 Experimental setup

**Data and training.** We train every model from scratch on 128M image-caption pairs sampled from DataComp (Gadre et al., 2024), filtered first by ImageNet-21K text overlap and then by a DFN (Fang et al., 2023); full details are in Appendix J. The batch size is 1500 across four RTX A6000 GPUs with mixed

Table 2: Zero-shot performance on the main benchmark suite. For each backbone, we compare a SigLIP baseline (HC=✗) with a matched HyperCLIP model (HC=✓). Bold marks the better of the pair. Classification: top-1 accuracy. Shifts: top-1 accuracy on ImageNet-R/O. Retrieval: top-1 mean recall on Flickr30k/MSCOCO. Fairness: worst-group top-1 accuracy on Dollar Street (DS) and GeoDE. Fine-tuning: top-1 accuracy after linear probing.

| Model | | Classification | | Shifts | | Retrieval | | Fairness | | Fine-tuning | |
| Arch. | HC | IN-1K | C100 | IN-R | IN-O | Flickr | COCO | DS | GeoDE | IN-1K | C100 |
|---|---|---|---|---|---|---|---|---|---|---|---|
| B0 | ✗ | 40.2 | 53.3 | 41.0 | 55.1 | 37.6 | 22.8 | 48.5 | 72.3 | 47.6 | 65.7 |
| B0 | ✓ | **42.6** | **55.0** | **44.6** | **57.0** | **41.2** | **24.7** | **50.0** | **72.7** | **50.1** | **66.9** |
| B1 | ✗ | 42.9 | 56.6 | 44.0 | 54.3 | 41.6 | 24.9 | 48.7 | **74.9** | 50.9 | 68.1 |
| B1 | ✓ | **45.1** | **57.9** | **47.8** | **55.6** | **43.9** | **26.6** | **49.1** | 74.6 | **53.2** | **69.0** |
| B2 | ✗ | 44.1 | 56.6 | 45.3 | 56.4 | 42.8 | 25.5 | 48.7 | 75.4 | 52.5 | 68.6 |
| B2 | ✓ | **46.6** | **59.1** | **50.5** | **57.7** | **45.9** | **28.4** | **52.2** | **75.5** | **55.0** | **70.1** |
| M0 | ✗ | 29.7 | 42.3 | 28.3 | 46.1 | 25.9 | 16.0 | 43.2 | 60.8 | 35.5 | 58.1 |
| M0 | ✓ | **32.6** | **47.9** | **32.4** | **49.5** | **29.1** | **17.6** | **44.5** | **64.2** | **37.7** | **60.6** |
| M1 | ✗ | 38.3 | 49.4 | 37.5 | 52.5 | 35.8 | 21.9 | 47.4 | 68.7 | 44.9 | 62.6 |
| M1 | ✓ | **40.3** | **52.6** | **40.4** | **54.7** | **37.0** | **23.1** | 47.4 | **71.1** | **46.9** | **64.9** |
| T0 | ✗ | 29.5 | 43.1 | 29.6 | 45.3 | 26.5 | 15.8 | 42.3 | 60.3 | 35.7 | 58.0 |
| T0 | ✓ | **32.8** | **46.4** | **33.1** | **50.3** | **29.2** | **17.5** | **43.9** | **63.2** | **38.2** | **59.4** |
| E0 | ✗ | 43.5 | **56.8** | 45.3 | **57.7** | 41.1 | 25.3 | 49.1 | 74.2 | 50.4 | **68.7** |
| E0 | ✓ | **44.6** | 55.9 | **47.6** | 57.3 | **43.3** | **26.7** | **51.4** | **75.0** | **51.9** | 66.5 |
| V0 | ✗ | 36.7 | 48.7 | 35.6 | **51.2** | 33.5 | 20.1 | **48.9** | 69.7 | 45.2 | 60.5 |
| V0 | ✓ | **37.7** | **50.6** | **36.8** | 50.4 | **35.6** | **20.9** | 48.2 | **70.4** | **45.7** | **60.6** |

precision. The hypernetwork takes as input the EOT-token text embedding for each caption, averaged within a mini-batch before the final feedforward layer.

**Architectures.** We evaluate eight backbones (Table 1); for each backbone we train both a SigLIP baseline and a HyperCLIP model with the hypernetwork of Section 3. The text encoder is identical across the two models.

**Evaluation.** We report zero-shot top-1 accuracy on ImageNet-1K and CIFAR-100 for classification; top-1 mean recall on Flickr30k and a 5K-image MSCOCO subset for retrieval; top-1 accuracy on ImageNet-R and ImageNet-O for distribution shift; and worst-group top-1 accuracy on Dollar Street and GeoDE for fairness. We additionally report top-1 accuracy on six further classification datasets (CIFAR-10, Caltech-101, Food101, Oxford-IIIT Pet, Pascal VOC 2007, STL-10) in Appendix A. Prompts come from the public OpenCLIP benchmark; full dataset details are in Appendix I.

**Linear probing.** For ImageNet-1K and CIFAR-100, we additionally fine-tune a linear classification head over the image encoder features. The head is initialized to the text-prompt embeddings; training uses AdamW (weight decay 0.1, learning rate $10^{-4}$) for 10 epochs on ImageNet-1K and 100 epochs on CIFAR-100.

### 4.2 HyperCLIP improves zero-shot accuracy across backbones

Table 2 shows the main result: HyperCLIP outperforms its matched SigLIP baseline on at least eight of the ten metrics for every BatchNorm-based backbone (B0–B2, M0–M1, T0). Quantitatively, HyperCLIP improves zero-shot ImageNet-1K accuracy by +2.4, +2.2, +2.5, +2.9, +2.0, and +3.3 percentage points for B0, B1, B2, M0, M1, and T0 respectively, and zero-shot CIFAR-100 accuracy by between +1.3 and +5.6 points. The two architectures that gain less, EdgeNext (E0, LayerNorm-only) and MobileViT (V0, mixed

BN+GN), are also the architectures with the fewest BatchNorm parameters for the hypernetwork to control; the trend is consistent with the BatchNorm-expressiveness picture of Frankle et al. (2020).

The size of the gains is enough to close gaps between adjacent model scales. A HyperCLIP-adapted EfficientNet-B0 reaches 42.6% on ImageNet-1K, within 0.3 points of the non-adapted EfficientNet-B1 (42.9%), which has 2.6M more parameters; the HyperCLIP-adapted B1 (45.1%) outperforms the non-adapted B2 (44.1%) despite having 1.2M fewer parameters. In other words, on these backbones, attaching a hypernetwork at training time is competitive with moving up one step on the EfficientNet scaling ladder, and adds nothing to the deployed inference cost.

The accuracy gains hold beyond zero-shot classification. HyperCLIP is at least as good as the SigLIP baseline on ImageNet-R and ImageNet-O distribution-shift benchmarks (Table 2, "Shifts"), on the Flickr30k and MSCOCO retrieval tasks ("Retrieval"), and on worst-group accuracy on Dollar Street and GeoDE ("Fairness"). Additional zero-shot classification results on six further datasets are in Appendix Table 3.

### 4.3   Linear probing

A natural objection is that any improvement from the hypernetwork might be recoverable by fine-tuning the matched baseline on labelled task data. We test this by additionally fine-tuning a linear classification head on ImageNet-1K and CIFAR-100 for both the HyperCLIP and SigLIP models. The two right-most columns of Table 2 show that the HyperCLIP advantage persists after linear probing: every HyperCLIP/baseline gap from the zero-shot column is preserved or widened. The hypernetwork captures information from the prompt embeddings that a linear probe over the frozen baseline encoder cannot recover, even with task-specific labels.

### 4.4   Scaling, throughput, and pruning

**Scaling with model size and training data.**   HyperCLIP's gain is preserved as we increase backbone size and training data. Sorting the BatchNorm-based backbones by parameter count (Appendix Table 5), the zero-shot improvement on CIFAR-100 (+1.3 to +5.6) and ImageNet-1K (+2.0 to +3.3) shows no systematic decay across the 1.7M–8.4M range we cover. Varying training-set size from 12.8M to 128M samples on EfficientNet-B0 (Appendix Table 6) yields ImageNet-1K gains in the 2.4–2.8 range and CIFAR-100 gains in the 2.0–3.8 range, again without a clear trend with scale. Scaling HyperCLIP to very large image encoders would require redesigning the hypernetwork output head and is left for future work (Appendix C).

**Training throughput.**   Training a HyperCLIP model is more expensive than training the matched baseline because we run the hypernetwork on every batch. The last row of Table 1 reports the relative training throughput. For most backbones the overhead is mild (−3% to −19%, and even modestly positive for M0, M1, V0 due to incidental batching effects). The exception is EdgeNext, where throughput drops by 48%; this is also the architecture with the smallest BatchNorm/LayerNorm budget and the smallest gain. LayerNorm's sequential statistics computation creates a parallelization bottleneck, and the small adaptation budget means there is correspondingly little benefit. The overhead is paid *only at training time*: at deployment, HyperCLIP and SigLIP run the same image encoder with the same inference cost. The training-time overhead is thus a one-time expense traded for a permanent, per-image-free accuracy gain. For most backbones the trade is favorable (overhead −3% to −19% for a +2 to +3 point zero-shot gain); the exception is EdgeNext, where a 48% throughput cost buys little gain, and we do not recommend HyperCLIP in that regime.

**Comparison with pruning.**   Pruning is an alternative way to obtain a small CLIP-style classifier from a larger one. We compare HyperCLIP-adapted EfficientNet-B1 (7.2M) with a pruned EfficientNet-B2 (8.4M, ∼14% of convolutional filters removed via PyTorch's L1-unstructured pruning). On CIFAR-100, pruning yields −0.6 points relative to the unpruned B2 baseline while HyperCLIP yields +2.0 points relative to the unadapted B1 baseline; on ImageNet-1K, the corresponding numbers are +0.8 vs +2.6 (Appendix Table 7). Even though the pruned model has more parameters, the HyperCLIP-adapted smaller model outperforms it on both benchmarks. Pruning also requires hardware support to translate parameter sparsity into latency improvements; HyperCLIP requires none.

# 5 Analysis: what the hypernetwork learns

The results in Section 4 establish that HyperCLIP improves performance, but they do not by themselves explain *why*. We argued in the introduction that the gain reflects something specific about the structure of CLIP-style models: a small image encoder is forced to solve an open-vocabulary problem it does not actually face at inference, and the class prompts are sufficient to relieve it of that burden through a small, well-localized control surface. This section assembles the evidence for that claim. We reuse the same experimental data as Section 4; the contribution here is interpretive rather than empirical. We state this interpretation as a hypothesis the evidence below is consistent with, not as a demonstrated mechanism. Controls (prompt perturbation and a capacity-matched variant; Appendix L) indicate that a substantial part of the measured gain is attributable to the hypernetwork's added training-time capacity and a task-agnostic normalization recalibration rather than to prompt-specific conditioning. We temper the causal language throughout this section accordingly and treat the precise mechanism as an open question.

## 5.1 The gain concentrates in BatchNorm-rich backbones

If BatchNorm scale and bias are the operative task channel, then backbones with a small BN budget should benefit less, and backbones whose normalization channel is non-BN should benefit least of all. Table 2 and Table 1 show exactly this pattern. The six BatchNorm-only backbones (B0, B1, B2, M0, M1, T0) all gain +2.0 to +3.3 points on ImageNet-1K and +1.3 to +5.6 points on CIFAR-100. EdgeNext (E0), which uses LayerNorm exclusively, gains +1.1 on ImageNet-1K and *loses* 0.9 on CIFAR-100, the only backbone where HyperCLIP underperforms the matched baseline on either of the two main classification benchmarks. MobileViT (V0), which has a mixed BN+GroupNorm budget but the smallest BN-only count of any BN backbone, gains the least among the BN architectures (+1.0 ImageNet-1K, +1.9 CIFAR-100). The same architectures are also the ones with the smallest number of adapted parameters (8.8K and 15.5K respectively, versus 42–68K for the BN-rich backbones; Table 1). This is the pattern one expects if the hypernetwork is specifically exploiting BatchNorm scale and bias as a task channel, and is not a generic adaptation mechanism that would apply equally to any small parameter group.

## 5.2 Class prompts recover half of supervised BatchNorm fine-tuning, with no labels

We can ask how close HyperCLIP gets to the best possible normalization-parameter adaptation. Given a matched SigLIP baseline and a labelled target task, the natural upper bound is to fine-tune the baseline's BatchNorm scale and bias *on that task* alongside the linear head. This bounds, from above, what an oracle method could extract from the BN channel. Figure 3 plots this upper bound, the SigLIP linear-probe baseline, and HyperCLIP on CIFAR-100 for the three EfficientNet backbones. Across B0–B2, the SigLIP upper bound is 3.34% above linear probing on average; HyperCLIP's zero-shot gain is 1.83%, roughly half. The significance of this result lies not in the absolute magnitude of 1.83%, but in the fact that it is recovered *from the class prompts alone*, without any task-specific labels, training, or gradient steps. The prompt embeddings carry enough task structure to recover a substantial portion of what supervised BN fine-tuning achieves. We caution that this recovery does not by itself establish that the *prompt content* is the operative signal: a capacity-matched, prompt-independent variant recovers a comparable portion (Appendix L), so part of the effect is a task-agnostic normalization adjustment. We are not aware of prior evidence that the text side of a contrastive VLM contains a usable task descriptor of this kind.

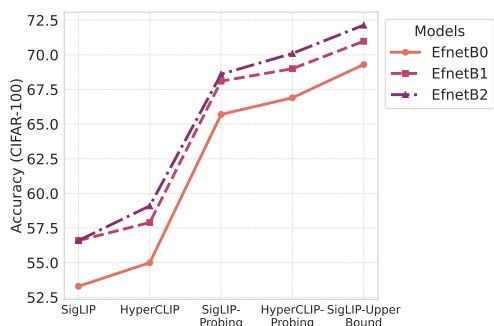

Figure 3: Class prompts vs supervised BatchNorm fine-tuning on CIFAR-100. "SigLIP Probing" is a linear probe over the frozen SigLIP image encoder. "SigLIP Upper Bound" additionally fine-tunes the encoder's BatchNorm scale and bias on the target task. HyperCLIP closes roughly half of this gap without seeing task-specific labels.

### 5.3 The gain is worth one step up the backbone scaling ladder

A useful way to size the gain is to ask how much extra backbone capacity it would take to obtain it the conventional way. EfficientNet provides a clean ladder: B0 (4.6M parameters) < B1 (7.2M) < B2 (8.4M). Reading off Table 2, HyperCLIP-adapted B0 reaches 42.6% on ImageNet-1K, 0.3 points below the non-adapted B1 (42.9%) which has 2.6M more parameters. HyperCLIP-adapted B1 reaches 45.1%, 1.0 point *above* the non-adapted B2 (44.1%) which has 1.2M more parameters. On this family, then, adding a hypernetwork at training time is worth roughly one step up the EfficientNet ladder. This is the magnitude of effect one would expect if a meaningful portion of the encoder capacity in the larger SigLIP models is spent on solving the open-vocabulary problem rather than on representing visual content: the hypernetwork allows the smaller backbone to avoid that part of the load.

### 5.4 Captions at training, class prompts at inference

A subtle but important property of HyperCLIP is that the input distribution of its hypernetwork shifts between training and inference. At training time, the hypernetwork sees web captions from DataComp: long, varied, noisy strings drawn from the data filtering pipeline. At inference time, it sees clean class-prompt embeddings such as *"a photo of a golden retriever"* (or the dataset-specific prompts from OpenCLIP). The hypernetwork is never trained directly on class prompts, yet its outputs on class prompts are useful enough that we observe the gains above. This means the hypernetwork has learned a function on the *embedding space* of the text encoder, not on the specific surface forms of any training caption. The text encoder is doing the work of mapping both captions and class prompts onto a shared embedding manifold, and the hypernetwork is acting on that manifold rather than on the strings underneath. We see this as additional evidence that the relevant task structure lives in the text-encoder representations themselves; HyperCLIP is making that structure visible through the BN channel. We quantify the caption-vs-prompt relationship in Appendix L (Table 12): class prompts occupy a tighter, more central sub-region of the caption-embedding space (mean pairwise cosine 0.44 vs 0.23; 97% of prompts lie within the caption support), consistent with a hypernetwork trained on captions transferring to prompts at inference.

### 5.5 Decomposing the hypernetwork: set aggregation versus deep representation

The hypernetwork has two distinguishable jobs. It must collapse a *set* of prompt embeddings into a single representation (set aggregation), and it must map that representation to a useful point in BN-parameter space (parameter generation). To isolate which of these the depth of the hypernetwork is doing, we replace the 12-layer Transformer with a single linear bottleneck layer mapping the average prompt embedding directly to $\Theta'$. Figure 4 shows the resulting drops for the EfficientNet family: on average, 1.7 points on CIFAR-100, with a worst case of 3.4 points on Dollar Street for B2. The simpler hypernetwork keeps the majority of HyperCLIP's gain in every case: aggregation plus a linear map is doing most of the work, and the Transformer adds a smaller refinement on top. Because a linear map from the mean prompt embedding captures most of the effect, we do not claim the 12-layer Transformer is the technically essential component; it is a refinement, and the near-linear variant is the more defensible default. This is a useful design point in its own right: when training-time cost matters, a near-linear hypernetwork is a viable substitute. It also speaks to the previous subsection's reading: if a linear map from the mean prompt embedding captures most of the structure, the structure must already be linearly accessible in the text-encoder embedding space.

## 6 Related work

We situate HyperCLIP among five neighbouring lines of work: hypernetworks as a parameter-generation mechanism, parameter-efficient adaptation of CLIP, the broader landscape of contrastive vision-language models, efficient on-device VLMs and VLM compression, and conditional computation including test-time adaptation.

**Hypernetworks for parameter generation.** HyperCLIP follows the classical hypernetwork paradigm (Schmidhuber, 1992; Ha et al., 2016; Denil et al., 2013; Bertinetto et al., 2016; Jia et al., 2016), in which one

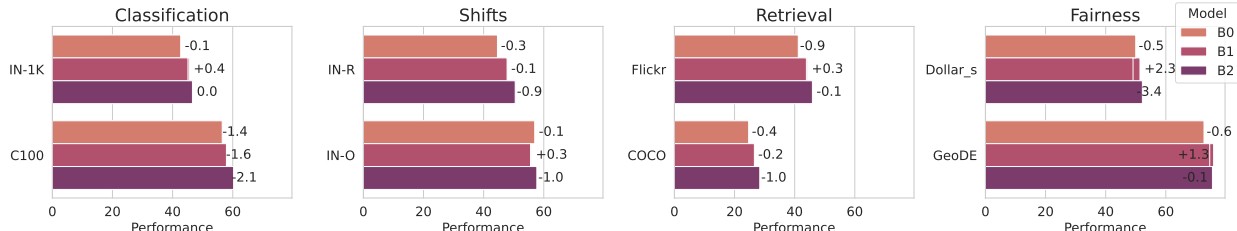

Figure 4: Decomposing the hypernetwork. Each bar is the performance drop on the EfficientNet family when the 12-layer Transformer is replaced by a single linear bottleneck mapping the mean prompt embedding to $\Theta'$. Classification: top-1 accuracy. Retrieval: top-1 mean recall. Fairness: worst-group top-1 accuracy. Most of HyperCLIP's gain survives the simplification: a linear map from the average prompt embedding does the bulk of the work, with the Transformer providing a 1–3 point refinement.

network generates the weights of another, and inherits the well-known difficulty of training such networks end-to-end without specialized initialization (Chang et al., 2023; Beck et al., 2023; Ortiz et al., 2023). Recent work has continued to extend this line in two directions relevant to us. First, hypernetworks have re-emerged as a way to amortize parameter-efficient fine-tuning: Charakorn et al. (2025) produce LoRA factors directly from a textual task description, and Moreno et al. (2024) use a single hypernetwork to generate LoRA and adapter weights across many sequence-labelling tasks. Second, Hedlin et al. (2024) introduce HyperNet Fields, which train hypernetworks without per-task ground-truth weights by learning along training trajectories. Outside language, hypernetworks are routinely used to specialize generative or editing models from a small conditioning signal, including text-to-image personalization (Ruiz et al., 2023) and StyleGAN inversion (Alaluf et al., 2022). HyperCLIP differs from these works in two respects: the conditioning signal is the full text-embedding set already produced by a CLIP-style text encoder, not a textual task description or a single image, and the generated parameters are normalization scale and bias rather than LoRA factors, a much smaller and more constrained adaptation surface that we find sufficient for end-to-end contrastive pre-training.

**Parameter-efficient adaptation of CLIP.** A large body of recent work adapts a frozen CLIP for a downstream task by attaching small trainable modules: low-rank weight updates on attention layers (Hu et al., 2021; Zanella & Ben Ayed, 2024a; Liu et al., 2024; Kopiczko et al., 2024), multi-modal adapters (Chen et al., 2022; Yang et al., 2024b), decoupled prompt-tuning channels (Zhang et al., 2024a), and unsupervised prompt distillation from a larger CLIP teacher (Li et al., 2024). These methods all share two design features: the inference-time vision encoder is the same size as the pre-trained one, and the adaptation itself requires per-task gradient descent on labelled or unlabelled target data. HyperCLIP differs from both. It treats adaptation as a function of the prompt set rather than as an optimization problem: a single forward pass through the hypernetwork produces a small specialized image encoder, with no per-task labels, no per-task gradient steps, and no added inference-time parameters relative to the matched SigLIP baseline.

**Contrastive VLMs and the SigLIP family.** HyperCLIP is built on top of the SigLIP sigmoid contrastive objective (Zhai et al., 2023), but the wider field has continued to refine contrastive losses and data pipelines. SigLIP 2 (Tschannen et al., 2025) integrates self-distillation and captioning auxiliaries; SuperClass (Huang et al., 2024) reframes contrastive pre-training as classification; Llip (Lavoie et al., 2024) explicitly models caption diversity; and Data Filtering Networks (Fang et al., 2024) together with DataComp (Gadre et al., 2024) drive the data-quality axis. In parallel, large-scale generalist VLMs such as InternVL (Chen et al., 2024), PaliGemma (Beyer et al., 2024; Steiner et al., 2024), and Qwen2-VL (Wang et al., 2024b) have pushed image-text capability at the multi-billion-parameter end. HyperCLIP is complementary to this entire axis: it does not introduce a new pre-training objective or dataset, and it would in principle compose with any of these losses or filtering procedures. Our claim is purely architectural: a small image encoder, when its normalization parameters are produced by a text-conditioned hypernetwork, is consistently better as a zero-shot classifier than the same encoder trained under the same objective without one.

**Efficient on-device VLMs and VLM compression.** The closest deployment-side work to HyperCLIP is the line of efficient CLIPs and small generalist VLMs. MobileCLIP (Vasu et al., 2024) and its follow-up MobileCLIP2 (Faghri et al., 2025) use multi-modal reinforced training to compress CLIP into a mobile-class image encoder; TinyCLIP (Wu et al., 2023) similarly trains a small student via affinity mimicking and weight inheritance; CLIP-KD (Yang et al., 2024a) systematically studies CLIP distillation; FastVLM (Vasu et al., 2025) attacks vision-token throughput rather than encoder size; MiniCPM-V (Yao et al., 2024) and SmolVLM (Marafioti et al., 2025) push generalist VLMs down to phone-class hardware. Orthogonally, model-level compression for VLMs targets distillation, structured pruning (Lin et al., 2024), post-training quantization (Wang et al., 2024a; Dettmers et al., 2022; Frantar & Alistarh, 2023), and general-purpose pruning approaches (Sun et al., 2023; Kuzmin et al., 2023; Liang et al., 2021; Yu et al., 2017; Han et al., 2016). HyperCLIP shares the goal of these works (a small image encoder for edge deployment) but takes a different trade-off: rather than compressing one image encoder that must serve every prompt set, it produces a different small encoder for each prompt set, paid for entirely at the moment the prompt set is known and absorbing no cost in the per-image forward pass.

**Conditional computation, mixture of experts, and test-time adaptation.** HyperCLIP can be read as input-conditional computation in which the conditioning signal is the full set of class-prompt embeddings and the conditioned quantity is the entire BatchNorm parametrization of the image encoder. This is a coarser modulation than per-feature FiLM-style conditioning (Perez et al., 2018; De Vries et al., 2017; Karras et al., 2019; Peebles & Xie, 2023) and a finer one than expert routing in CLIP-MoE variants such as MoDE (Ma et al., 2024), CLIP-MoE (Zhang et al., 2025), and MoE adapters for continual VLM learning (Yu et al., 2024), which select among a discrete set of cluster- or task-specific experts. A separate line of test-time adaptation methods also specializes CLIP at inference, but typically by maintaining a feature cache (Karmanov et al., 2024; Zhang et al., 2024b), by gradient-tuning prompts on each test sample (Yoon et al., 2024; Xiao et al., 2025), by training-free augmentation (Zanella & Ben Ayed, 2024b; Farina et al., 2024), or by optimal-transport reweighting (Zhang et al., 2024c). HyperCLIP commits adaptation up front, at the moment the label set is known, and removes it from the per-image forward pass entirely.

## 7 Conclusion

The central claim of this paper is not that adding a hypernetwork to CLIP improves zero-shot accuracy; it is that doing so makes a previously implicit piece of structure in contrastive vision-language models visible and usable. CLIP-style image encoders are trained to discriminate every category set, but at inference they only need to discriminate one, and the prompts that specify which one are already present in the model, encoded by the text encoder. HyperCLIP shows that this latent task descriptor is accessible: a small hypernetwork can recover it from the prompt embeddings and use it to re-tune a narrow, well-localized control surface in the image encoder (BatchNorm scale and bias). The empirical signature of this view is consistent across our experiments. The gain concentrates in BatchNorm-rich backbones and disappears in LayerNorm-only ones, consistent with BN being the operative channel. The magnitude of the gain is equivalent to one step up the EfficientNet scaling ladder, consistent with the small encoder being relieved of part of an open-vocabulary requirement it should not have had to satisfy. And against a supervised upper bound that fine-tunes BatchNorm on each downstream task, HyperCLIP recovers roughly half (without any task labels), consistent with the prompt embeddings already containing the relevant task structure. We present this as the interpretation most consistent with the evidence, not as a settled mechanism. Controls that separate prompt content from added capacity (Appendix L) show that a meaningful share of the gain is a capacity or regularization effect together with a task-agnostic normalization recalibration; how much of the remainder is genuinely prompt-conditioned is an open question we leave for future work. The empirical improvement itself is robust and reproduces across seeds.

Several questions follow naturally from this framing. First, BatchNorm is a useful but narrow control surface; if the prompt-conditioned task descriptor is real, it should also drive richer adaptations (low-rank updates to convolution weights, mixture-of-experts routing, attention re-weighting), provided hypernetwork training dynamics for those larger output spaces can be controlled. Second, the hypernetwork acts on text-encoder *embeddings* rather than on prompt strings, so the same machinery should compose with any contrastive

VLM that produces those embeddings; SigLIP is convenient but incidental. Third, the present work uses class prompts as the task descriptor, but other inference-time signals (a small image set, a region of interest, a user-supplied attribute list) would route through the same architecture and merit comparison. If the broader claim is right, the right question is not how much we can shrink one image encoder, but how much of contrastive pre-training's apparent scale requirement is downstream of an open-vocabulary assumption that the inference setting does not actually impose.

### Broader Impact Statement

HyperCLIP reduces the cost of deploying contrastive vision-language classifiers in resource-constrained settings, which lowers the barrier to using such models on edge devices. The downstream consequences (positive and negative) are inherited from those of CLIP-style models more generally: HyperCLIP does not change the training data or the loss, only the architecture, and so its risks and mitigations are those of the underlying contrastive pre-training paradigm. Because HyperCLIP adjusts the image encoder as a function of the label set, its behavior, including any bias, can in principle vary across label sets in ways a fixed encoder does not. Our fairness evaluations (Dollar Street, GeoDE) do not fully capture this label-set-dependent variation, and we flag it as a concrete direction for future auditing.

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

## A  Additional zero-shot results

We evaluate on classification tasks with test sets of ImageNet-1K (IN-1K), CIFAR-100 (C100), CIFAR-10 (C10), Caltech-101 (Ca101), Food101 (F101), Oxford-IIIT Pet (Pet), Pascal VOC 2007 (VOC), and STL-10, reporting top-1 zero-shot accuracy. Retrieval is evaluated on Flickr30k and a subset of MSCOCO 2014, reporting top-1 mean recall. Distribution-shift datasets are ImageNet-R (renditions of 200 ImageNet classes) and ImageNet-O (examples chosen because they are misclassified with high confidence by a ResNet-50). Fairness datasets are GeoDE and Dollar Street; we report worst-group top-1 zero-shot accuracy. All evaluation prompts come from the public OpenCLIP benchmark.

Table 3: Zero-shot top-1 accuracy on six additional classification datasets. "Arch" is the image encoder architecture; "HC" marks the experiments using HyperCLIP.

| Arch | HC | Ca101 | C10 | F101 | Pet | VOC | STL-10 |
|------|-----|-------|-----|------|-----|-----|--------|
| B0 | ✗ | 76.2 | 80.9 | 48.9 | 57.0 | 60.3 | 88.3 |
| B0 | ✓ | **78.9** | **82.8** | **51.4** | **60.3** | **63.1** | **89.0** |
| B1 | ✗ | 78.2 | **84.4** | 52.5 | 62.5 | **65.1** | 88.7 |
| B1 | ✓ | 78.2 | 84.0 | **55.0** | **63.4** | 63.4 | **89.9** |
| B2 | ✗ | 78.5 | 84.5 | 53.8 | 63.5 | **66.4** | 90.5 |
| B2 | ✓ | **81.1** | **85.4** | **56.5** | **66.0** | 65.4 | **90.6** |
| M0 | ✗ | 67.5 | 68.3 | 37.2 | 47.5 | 52.8 | 75.2 |
| M0 | ✓ | **71.2** | **74.1** | **39.2** | **53.3** | **55.0** | **80.7** |
| M1 | ✗ | 74.2 | 78.4 | 46.4 | 56.8 | 62.5 | 86.4 |
| M1 | ✓ | **76.4** | **80.8** | **49.2** | **57.0** | **63.1** | 86.4 |
| T0 | ✗ | 69.2 | 69.1 | 35.6 | 47.9 | 42.6 | 78.1 |
| T0 | ✓ | **73.6** | **74.0** | **39.4** | **51.4** | **53.1** | **80.1** |
| E0 | ✗ | **80.1** | **84.9** | 52.6 | 60.5 | 62.9 | 90.2 |
| E0 | ✓ | 79.0 | 84.3 | **54.3** | **63.5** | **65.4** | **90.7** |
| V0 | ✗ | **71.9** | 75.4 | 47.7 | 51.4 | **59.8** | 86.1 |
| V0 | ✓ | 71.6 | **78.5** | **48.9** | **55.1** | 59.6 | **87.6** |

# B  Ablation: training batch size

We vary the training batch size for EfficientNet-B0 to confirm that HyperCLIP's gains are not specific to a particular batch size. Table 4 shows that HyperCLIP outperforms the matched SigLIP baseline at every batch size we tried.

Table 4: Batch-size ablation on EfficientNet-B0. For each batch size, the upper row is the SigLIP baseline and the lower row is HyperCLIP.

| Batch | IN-1K | C100 | IN-R | IN-O | Flickr | COCO | DS | GeoDE |
|-------|-------|------|------|------|--------|------|-----|-------|
| 500 baseline | 36.4 | 51.3 | 38.2 | 50.9 | 34.7 | 20.3 | 48.8 | 71.6 |
| 500 HyperCLIP | 38.7 | 53.9 | 41.4 | 53.5 | 37.6 | 21.9 | 49.1 | 72.4 |
| 700 baseline | 38.0 | 53.1 | 39.3 | 53.6 | 35.3 | 21.7 | 49.2 | 71.9 |
| 700 HyperCLIP | 39.7 | 53.8 | 42.9 | 53.4 | 38.3 | 22.8 | 49.5 | 72.4 |
| 1000 baseline | 39.4 | 52.0 | 40.3 | 53.9 | 36.6 | 22.1 | 49.3 | 70.7 |
| 1000 HyperCLIP | 41.7 | 55.1 | 44.2 | 54.6 | 38.6 | 24.1 | 49.3 | 73.2 |
| 1700 baseline | 40.4 | 53.1 | 40.8 | 54.5 | 37.6 | 22.7 | 48.6 | 69.9 |
| 1700 HyperCLIP | 42.9 | 55.6 | 44.7 | 56.5 | 41.1 | 25.3 | 50.5 | 73.9 |

# C  Scalability of HyperCLIP

HyperCLIP is targeted at the small-model regime: our motivation is edge deployment, where the image encoder must be small. We characterize its scaling along two axes within that regime.

**Model size.**  As the image encoder grows, the output dimension of the hypernetwork grows in proportion to the number of normalization parameters. Table 5 shows HyperCLIP's gain over its matched SigLIP baseline, sorted by backbone size, for the BatchNorm-based architectures. Gains are roughly constant across the 1.7M–8.4M parameter range we cover.

Table 5: HyperCLIP improvement over the matched SigLIP baseline, sorted by image encoder size.

| Model | Params (M) | CIFAR-100 (%) | ImageNet-1K (%) |
|-------|-----------|---------------|-----------------|
| T0 | 1.7 | +3.3 | +3.3 |
| M1 | 2.0 | +3.2 | +2.0 |
| B0 | 4.6 | +1.7 | +2.4 |
| M0 | 4.9 | +5.6 | +2.9 |
| B1 | 7.2 | +1.3 | +2.2 |
| B2 | 8.4 | +2.5 | +2.5 |

**Training-set size.** Table 6 reports HyperCLIP's gain on EfficientNet-B0 as we vary the number of training samples from 12.8M to 128M. The improvement is preserved at every scale.

Table 6: HyperCLIP-EfNetB0 gain over the matched SigLIP baseline as training sample count varies.

| Samples | CIFAR-100 (%) | ImageNet-1K (%) |
|---------|---------------|-----------------|
| 12.8M | +2.2 | +2.5 |
| 51.2M | +3.8 | +2.6 |
| 102.4M | +2.9 | +2.8 |
| 128.0M | +2.0 | +2.4 |

## D  Comparison with pruning

We compare HyperCLIP-adapted EfficientNet-B1 (7.2M parameters) with a pruned EfficientNet-B2 (8.4M, with $\sim 14\%$ of convolutional filters pruned via PyTorch's built-in L1-unstructured pruning). On both CIFAR-100 and ImageNet-1K the HyperCLIP-adapted smaller model improves more over its un-modified baseline than pruning does over its un-modified baseline (Table 7). We also note that pruning typically requires specialized hardware to translate parameter sparsity into latency improvements, whereas HyperCLIP needs none.

Table 7: HyperCLIP vs. pruning. Each entry is the change in zero-shot accuracy relative to the un-modified backbone of the same size.

| Model | Params | CIFAR-100 ($\Delta\%$) | ImageNet-1K ($\Delta\%$) |
|-------|--------|------------------------|---------------------------|
| EfNetB2, 14% pruned | 8.4M | −0.6 | +0.8 |
| EfNetB1 + HyperCLIP | 7.2M | +2.0 | +2.6 |

## E  Optional weight-scale parameter

To diagnose the training dynamics of contrastive losses with and without a hypernetwork, we run a controlled experiment on synthetic data with two small linear-plus-BatchNorm networks, ModelX and ModelY, trained under CLIP, SigLIP, and SigLIP+hypernetwork over 100 epochs. Figure 5 shows the evolution of parameter norms (left) and update norms (right). Adding a hypernetwork changes the trajectory of parameter norms early in training and then stabilizes them.

This observation motivates an optional learnable weight-scale parameter $S_w$, applied to the output of $FF_{\text{output}}$. To set $S_w$, we briefly train SigLIP and HyperCLIP for a small number of steps (e.g., 1M samples), measure the norm of the adapted BatchNorm parameters in SigLIP, and set $S_w$ so that the hypernetwork output matches that scale at the start of training. With this initialization, the optional weight-scale parameter adds a small further improvement: +0.15, +0.68, and +0.45 points on ImageNet-1K for B0, B1, and T0 respectively.

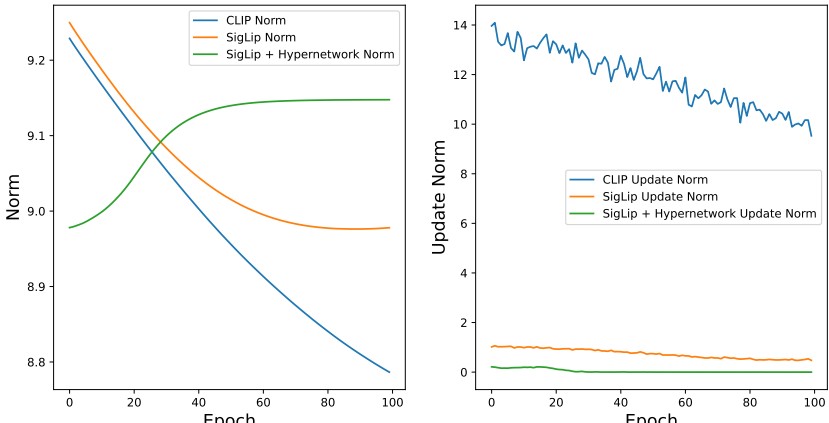

Figure 5: Parameter norms (left) and update norms (right) over 100 training epochs for CLIP, SigLIP, and SigLIP+hypernetwork on a synthetic two-network controlled experiment. The hypernetwork variant stabilizes the parameter-norm trajectory after an initial transient.

## F  Hypernetwork hyperparameters

The hypernetwork consists of an input projection $FF_{\text{input}}$, a 12-layer non-causal Transformer encoder (width 768, 8 heads, FFN dimension 2560, GELU activation, dropout 0.1), a bottleneck linear layer, a LayerNorm, and an output feedforward layer $FF_{\text{output}}$ whose output dimensionality equals $|\Theta'|$. Each prompt embedding is a single token to the Transformer; after the Transformer, we take the per-token mean before the bottleneck. Bottleneck dimensions are listed in Table 1.

All SigLIP and HyperCLIP models are trained from scratch on 128M filtered DataComp samples for one epoch. Linear probing is run for 10 epochs on ImageNet-1K and 100 epochs on CIFAR-100, optimizing cross-entropy with AdamW (weight decay 0.1, learning rate $10^{-4}$).

## G  Predicting non-normalization parameters

We tried extending the hypernetwork to predict additional parameter groups (linear layer weights and convolutional filters), and found that these variants did not outperform the SigLIP baseline. We attribute this to two factors: (i) those parameter groups are much larger, leading to an unwieldy output head whose dynamics are difficult to control, and (ii) they live in a part of the network where standard initialization assumptions matter more, and the hypernetwork output does not respect those assumptions out of the box. Table 8 reports the corresponding zero-shot numbers on EfficientNet-B0.

Table 8: Predicted parameter group on EfficientNet-B0, as $\Delta$ zero-shot top-1 over the matched SigLIP baseline (in-run; baseline = 0). Extending the hypernetwork to convolutional or linear-layer weights does not beat the BatchNorm target or the baseline. Richer targets are run at a 32M-sample budget.

| Predicted $\Theta'$ | $\Delta$ IN-1K | $\Delta$ C100 |
|---|---|---|
| BatchNorm scale + bias [ours] | +2.0 | +0.9 |
| + linear weights (32M) | −16.4 | −36.8 |
| + convolutional filters (32M) | −12.6 | −52.4 |

Within the normalization choices, LayerNorm-based backbones gain less than BatchNorm-based ones (E0 in Table 2). Two reasons: LayerNorm typically has fewer parameters in these architectures (EdgeNext has roughly 7× fewer normalization parameters than EfficientNet-B1 at comparable total parameter count), and

LayerNorm statistics vary less across examples, leaving less for dynamic prediction to exploit. Predicting larger subsets of the encoder while keeping training stable is the most natural extension of HyperCLIP and the most likely source of further gains.

## H   HyperCLIP for dense prediction tasks

HyperCLIP is agnostic to the downstream use of the image encoder. Dense-prediction pipelines that build on top of a frozen CLIP image encoder by attaching a task-specific head can take a HyperCLIP-adapted encoder unchanged: the gains we observe under linear probing (Section 4.3) suggest the adapted features remain useful when used as a frozen backbone.

## I   Training libraries, datasets, and metrics

We use OpenCLIP (Ilharco et al., 2021) and Timm (Wightman, 2019) where applicable. Augmentation is restricted to random resized crop and normalization. Training uses batch size 1500 on four RTX A6000 GPUs with mixed precision. For throughput measurements we find the maximum batch size that fits on one A100 without OOM or exceeding PyTorch's int32 indexing limits, and report relative changes. Datasets and sizes are in Table 9.

Table 9: Evaluation datasets.

| Dataset | Description | Test size |
|---------|-------------|-----------|
| ImageNet-1K (IN-1K) | 1,000 object classes | 50,000 images |
| CIFAR-100 (C100) | 100 classes, 100 images/class | 10,000 images |
| CIFAR-10 (C10) | 10 classes, 1,000 images/class | 10,000 images |
| Food101 (F101) | 101 food classes, 250 images/class | 25,250 images |
| Oxford-IIIT Pet | 37 pet classes | 3,669 images |
| Pascal VOC 2007 | 20 object classes | 14,976 images |
| STL-10 | 10 classes | 8,000 images |
| Flickr30k | Image-caption retrieval | 1,000 images, 5,000 captions |
| MSCOCO 2014 | Image-caption retrieval | 5,000 images |
| ImageNet-R (IN-R) | 200 classes, robustness eval | 30,000 images |
| ImageNet-O (IN-O) | 200 classes, OOD eval | 2,000 images |
| GeoDE | 40 classes, geographic-diversity eval | 12,488 images |
| Dollar Street | 58 classes, socioeconomic-diversity eval | 3,503 images |

**Recall@1.**   For retrieval, image-retrieval recall@1 averages over captions the indicator that at least one correct image appears in the top-1 results: $\frac{1}{N}\sum_{i=1}^{N}\mathbb{1}[\text{recall@1}_i > 0]$, where $N$ is the number of captions. Text-retrieval recall@1 is defined analogously over images. Top-1 mean recall is the average of the two.

## J   DataComp with data filtering networks

Following Gadre et al. (2024), we sample image-caption pairs from DataComp pools and apply two stages of filtering. First, we use FastText (Bojanowski et al., 2017) to keep English captions and intersect with ImageNet-21K synsets, leaving captions whose text overlaps with ImageNet class names. Second, we apply DFN-based filtering (Fang et al., 2023), keeping image-caption pairs that pass the public DFN trained on HQITP-350M. The result is approximately 100M unique pairs; we sample 128M with replacement for one epoch of training. The filtering pipeline is shared between every HyperCLIP model and its matched SigLIP baseline, so any HyperCLIP gains we report are not attributable to data filtering.

## K   Properties of sigmoid training

SigLIP (Zhai et al., 2023) uses a sigmoid loss instead of the softmax cross-entropy used by CLIP. Three properties make it convenient for our setting. First, it is more robust to label noise because each pair contributes an independent term. Second, it is memory-efficient at large batch sizes: a softmax-based loss materializes a $|B| \times |B|$ pairwise-similarity matrix, while SigLIP can be chunked to a per-device $b \times b$ matrix and summed across devices, reducing peak memory from $O(|B|^2)$ to $O(b^2)$. Third, it remains effective at small batch sizes, where softmax-based CLIP is comparatively weak. HyperCLIP's contribution is orthogonal to the choice of contrastive loss; we use SigLIP because it is the current state-of-the-art baseline, but the same hypernetwork attachment would apply to a softmax-based CLIP.

**Relationship to LoRA.**   LoRA (Hu et al., 2021) and HyperCLIP both adapt a model with few parameters, but operate at different points in the pipeline. LoRA introduces additional trainable low-rank matrices that are fit *after* pre-training on a per-task basis; the adapted parameters are stored separately for each task. HyperCLIP adapts a single jointly trained network at inference time using only the text prompts as input, with no per-task fitting. LoRA is also typically applied to large language models, while HyperCLIP targets small vision encoders for edge deployment.

## L   Controls and additional results

This appendix collects control experiments that isolate the source of HyperCLIP's gain. All are on EfficientNet-B0, using models trained under the same pipeline as the main experiments. To keep every comparison matched, we report results as within-run deltas over the relevant baseline rather than as absolute accuracies. Controls run at a reduced data budget are marked; those comparisons are matched within budget.

**Prompt-perturbation controls.**   We keep the trained HyperCLIP model fixed and, at inference only, replace the class prompts fed to the hypernetwork with (a) shuffled prompts, (b) unrelated random text, and (c) a mismatched label set. The classifier head still uses the true prompts; only the hypernetwork's conditioning input is perturbed. Table 10 shows the gain over the SigLIP baseline is essentially unchanged under every perturbation, rather than collapsing toward zero: the trained model is close to invariant to prompt content.

Table 10: Prompt-perturbation control on EfficientNet-B0, as $\Delta$ zero-shot top-1 over the matched SigLIP baseline (in-run; baseline = 0). Perturbing the prompt content fed to the hypernetwork retains essentially the full HyperCLIP gain over the baseline, rather than reverting to it.

| Conditioning of $\mathcal{H}$ | $\Delta$ IN-1K | $\Delta$ C100 |
|---|---|---|
| HyperCLIP, true prompts | +2.0 | +0.9 |
| shuffled prompts | +2.0 | +0.8 |
| unrelated random text | +2.0 | +0.8 |
| mismatched label set | +2.0 | +0.8 |

**Capacity-matched control.**   To separate the hypernetwork's added capacity from prompt conditioning, we replace the prompt embedding with a single learned, prompt-independent token, keeping the rest of the hypernetwork and the optimization path unchanged. At a matched budget this prompt-independent variant matches the true-prompt HyperCLIP to within 0.2 points on ImageNet-1K, i.e. removing the prompt does not remove the improvement. Consistent with this, the trained hypernetwork's input projection is driven to near-zero and its output is close to invariant across prompts (cosine similarity $\approx 1$ between outputs for different label sets), which also explains the prompt-perturbation result above.

**Multi-seed results.**   Table 11 reports mean $\pm$ s.d. over seeds for the B0 main comparison. The HyperCLIP–SigLIP gap ($\approx +2.6$ on ImageNet-1K) is much larger than the per-model seed spread (0.1 to 0.4), so the improvement is not seed noise.

Table 11: Multi-seed B0 main comparison, as the HyperCLIP − SigLIP gap in zero-shot top-1 (mean ± s.d. over seeds). The gap is roughly eight times the per-model seed spread (0.1 to 0.4), so it is not seed noise.

| Comparison | Δ IN-1K | Δ C100 |
|---|---|---|
| HyperCLIP − SigLIP | $+2.6 \pm 0.3$ | $+2.1 \pm 0.4$ |

**Caption-vs-prompt embedding statistics.** Table 12 quantifies the train/inference text distribution shift discussed in Section 5.4. Class prompts are more mutually similar and more central than captions, and almost all prompts fall within the caption support, consistent with a hypernetwork trained on captions transferring to prompts at inference.

Table 12: Caption vs class-prompt embedding statistics (text-encoder embedding space). Class prompts are more mutually similar and more central than training captions.

| Metric | Captions | Prompts |
|---|---|---|
| Mean pairwise cosine | 0.227 | 0.440 |
| Mean cosine to caption centroid | 0.479 | 0.526 |
| Fraction of prompts inside caption support | n/a | 0.973 |

**Post-hoc adaptation baseline.** Table 13 fits the hypernetwork on top of a frozen, already-trained SigLIP encoder (8M-sample budget) rather than training jointly. It does not improve over the frozen baseline, indicating that whatever benefit the hypernetwork provides arises during joint pre-training rather than from a post-hoc prompt-to-normalization map.

Table 13: Post-hoc hypernetwork fit on a frozen SigLIP-B0 encoder (8M-sample budget), as Δ zero-shot top-1 over the frozen baseline. Fitting the prompt-to-normalization map after pre-training does not improve over the frozen baseline.

| | Δ IN-1K | Δ C100 |
|---|---|---|
| Post-hoc hypernetwork (frozen encoder) | −0.1 | −0.1 |

**Summary.** Taken together, these controls indicate that HyperCLIP's improvement is robust (it reproduces across seeds) but that its source is not cleanly attributable to prompt-specific conditioning. Perturbing prompt content leaves accuracy at the HyperCLIP level, and a capacity-matched prompt-independent variant matches the true-prompt model at a fixed budget; both point to the hypernetwork's added training-time capacity, together with a task-agnostic normalization recalibration, as a substantial part of the effect. We accordingly temper the mechanistic language in Sections 1, 5, and 7, and present prompt-conditioning as a hypothesis consistent with the aggregate evidence rather than a demonstrated mechanism. Quantifying how much of the residual gain is genuinely prompt-conditioned is left for future work.

