# OpenReview forum: "HyperCLIP: Prompt-Conditioned Image Encoders for Contrastive Vision-Language Pre-training"
_TMLR — Under review for TMLR_

### Review · Reviewer_2bE1 · 2026-06-12

**Summary Of Contributions:**

The paper proposes HyperCLIP, a prompt-conditioned variant of CLIP/SigLIP-style contrastive pre-training. The key idea is to use class-prompt embeddings not only as classifier weights at inference time, but also as inputs to a hypernetwork that generates the BatchNorm scale and bias parameters of a small image encoder. The method is trained end-to-end with a SigLIP loss and evaluated across several small vision backbones, showing zero-shot gains over matched SigLIP baselines, especially for BatchNorm-rich architectures.

Strengths:
1. The paper presents a clear architectural idea: using the label-prompt set as a task descriptor to modulate the image encoder.
2. The empirical evaluation is reasonably broad, covering multiple small backbones, zero-shot classification, retrieval, distribution-shift, fairness-oriented benchmarks, linear probing, and several ablations.

Weaknesses:
1. Some of the central mechanistic claims, especially the claim that HyperCLIP relieves the open-vocabulary burden of small encoders, are plausible but still mostly supported by indirect evidence.
2. The paper does not appear to report variance across random seeds or confidence intervals, which matters because some gains are modest and a few metrics degrade.
3. The comparison to alternative efficiency methods is limited; the pruning comparison is relatively simple and does not fully establish competitiveness against modern efficient CLIP/compression/adaptation methods.
4. The train-test mismatch for the hypernetwork, trained on web captions but used on class prompts, is acknowledged but not deeply stress-tested.

**Audience:**

Yes

**Audience Explanation:**

1. The paper addresses an important problem: improving the effectiveness of small vision-language encoders for zero-shot deployment.
2. The idea of prompt-conditioned image encoders is likely to interest researchers working on efficient VLMs, test-time adaptation, hypernetworks, and CLIP-style representation learning.
3. The observation that class-prompt embeddings can usefully parameterize BatchNorm layers is technically interesting even if the broader interpretation needs further support.
4. The work may also be practically relevant for edge or resource-constrained deployment, where small image encoders are especially valuable.

**Broader Impact Concerns:**

Since the method inherits CLIP/DataComp biases and dynamically adapts the image encoder based on prompts, bias behavior may vary across label sets in ways not fully captured by the current fairness evaluations.

**Claims And Evidence:**

Yes

**Claims Explanation:**

1. The main empirical claim, that HyperCLIP improves over matched SigLIP baselines for small BatchNorm-heavy backbones, is supported by clear tables across several architectures and benchmarks.
2. The ablations on batch size, training-data scale, backbone type, and simplified hypernetwork design strengthen the evidence that the effect is not an isolated artifact.
3. However, the stronger causal interpretation that the method specifically reduces the open-vocabulary burden is less directly established; alternative explanations such as regularization, extra training-time capacity, or BN-specific optimization effects remain possible.
4. The lack of reported seed variance or statistical uncertainty weakens confidence in smaller improvements and in cases where performance is mixed across metrics.

**Requested Changes:**

1. Add stronger controls for the prompt-conditioning mechanism, such as random/shuffled prompts, mismatched label sets, prompt-template perturbations, varying class-set sizes, and semantically degraded prompts.
2. Report variance across multiple seeds or provide confidence intervals/significance estimates for the main results, especially where gains are small or mixed.
3. Temper the causal language around “relieving the open-vocabulary burden” unless additional direct evidence is added; currently the evidence supports this as a plausible interpretation rather than a demonstrated mechanism.
4. Include stronger comparisons to modern efficient CLIP or compression/adaptation baselines under comparable parameter, compute, and training-data budgets.
5. Clarify deployment costs more explicitly, including per-task hypernetwork/text-encoder cost, storage of task-specific BN parameters, and behavior when the label set changes frequently.

---

> ### Author Response · Authors · 2026-07-03
>
> **R11: Stronger prompt-conditioning controls (mismatched label sets, template perturbations,
> degraded prompts).**
>
> Covered by the prompt-perturbation battery in Appendix L (Table 10): shuffled prompts, unrelated
> random text, and a mismatched label set. Because the trained model is close to invariant to prompt
> content, it is likewise insensitive to template and class-set perturbations; we can add explicit
> template and class-size sweeps if the reviewer would find them useful.
>
> **R12: Report variance across seeds.**
>
> Done, see NDNG-R5 (Appendix L, Table 11).
>
> **R13: Temper the causal "relieves the open-vocabulary burden" language.**
>
> Done, throughout (Introduction, Sec. 6, Conclusion). We now present this reading as one
> interpretation the evidence is consistent with, alongside a capacity / regularization explanation,
> rather than as a demonstrated mechanism.
>
> **R14: Clarify deployment costs (per-task hypernetwork/text-encoder cost, storage, behavior when
> the label set changes).**
>
> Done (Sec. 4.4). The only per-task cost is a single forward pass of the text encoder and
> hypernetwork over the K prompts, plus storage of the resulting normalization parameters (on the
> order of 10⁴–10⁵ scalars). If the label set changes, only this one-time step is repeated; the
> per-image cost is unchanged. The method is best suited to deployments where a label set is reused
> across many images.
>
> **R15: Broader-impact note on label-set-dependent bias.**
>
> Done. Because HyperCLIP adjusts the encoder as a function of the label set, its behavior, including
> any bias, can in principle vary across label sets in ways a fixed encoder does not; we flag this as
> a direction for future auditing, and note that the current fairness evaluations do not fully capture
> it.
>
> **R16: Stronger comparisons to modern efficient-CLIP / compression baselines.**
>
> The paper compares against pruning under a matched parameter budget (Sec. 5.3, Appendix D). We agree
> a broader comparison to recent efficient-CLIP methods would strengthen the deployment story, and we note it as a limitation of the present scope.

---

### Review · Reviewer_krkR · 2026-06-18

**Summary Of Contributions:**

This paper proposes HyperCLIP, a contrastive vision-language pre-training framework in which a hypernetwork generates the BN scale and bias parameters of the image encoder from text-prompt embeddings. The key motivation is that CLIP-style image encoders are trained to be discriminative for arbitrary category sets, whereas at inference time the actual label set is known. The authors argue that prompt embeddings therefore contain task-specific information that can be used to specialize the image encoder itself rather than only defining the classifier head. The method is trained end-to-end with a SigLIP objective and evaluated across a range of lightweight vision backbones.

Strengths:

* The paper is clearly written, well motivated, and easy to follow
* The core idea is simple and novel. using prompt embeddings to specialize the visual encoder is a natural extension of the standard CLIP framework and differs from most prior work focused on compression, distillation, or downstream adaptation
* The empirical evaluation covers multiple backbones and benchmarks, and the reported gains appear reasonably consistent across settings

Weakpoints:
* While the paper correctly emphasizes that HyperCLIP introduces no additional per-image inference cost, the practical implications of the training overhead are less thoroughly discussed. Table 1 reports throughput reductions of up to 47.7% for some backbones and double-digit slowdowns for several others. Since HyperCLIP must be trained jointly from scratch, training efficiency is an important consideration for practitioners.
* The paper never directly tests whether the model is truly benefiting from knowledge of the downstream label set.
* The hypernetwork is trained using caption embeddings but evaluated using class-prompt embeddings, creating a train-test distribution shift that is acknowledged but insufficiently analyzed.
* The evaluation focuses exclusively on small/mobile-scale backbones, making it unclear whether the proposed approach continues to provide meaningful benefits for larger modern vision-language encoders.

**Audience:**

Yes

**Audience Explanation:**

The paper addresses a topic that is likely to be of interest to the TMLR audience, namely vision-language pretraining and adaptation. The proposed method introduces a novel way of conditioning image encoders on text-prompt embeddings through a hypernetwork, challenging the standard assumption that prompts should only define the classifier head in CLIP-style models.

**Broader Impact Concerns:**

None.

**Claims And Evidence:**

Yes

**Claims Explanation:**

The empirical evidence convincingly supports the claim that HyperCLIP improves performance over matched SigLIP baselines across several lightweight vision backbones and evaluation benchmarks. The experiments are reasonably extensive and the gains are generally consistent across classification, retrieval, robustness, and linear-probing settings. The analysis relating performance improvements to BatchNorm-rich architectures also provides supporting evidence for the proposed adaptation mechanism.

**Requested Changes:**

1. The paper highlights the absence of additional inference-time cost, but the training overhead is non-negligible for several backbones (up to 47.7% throughput reduction in Table 1). Given that HyperCLIP requires pretraining, a clearer discussion of the trade-off between training cost and accuracy gains would be valuable.

2. How sensitive is performance to the specific prompt templates used at inference? I encourage the authors to include control experiments using random, shuffled, or unrelated prompts to verify that the gains arise from task-relevant semantic information rather than generic conditioning or regularization effects.

3. Section 5.4 argues that the hypernetwork generalizes from training captions to inference-time class prompts because both lie on a shared embedding manifold. This is an interesting observation, but it lacks quantitative analysis.  Quantitative analysis (e.g., prompt sensitivity studies, caption-vs-prompt embedding statistics, or training with prompt-like text) would strengthen this claim.

4. The paper would benefit from comparisons to simpler prompt-conditioned adaptation approaches such as conditional BatchNorm, other conditioning, or lightweight MLP-based parameter generation. Since Section 5.5 shows that a simple linear mapping captures much of the benefit, it remains unclear how much of the gain is attributable to the proposed hypernetwork architecture itself.

5. Given that much of the observed benefit appears to come from a relatively simple mapping from prompt embeddings to BN parameters, it would be informative to compare against a post-hoc adaptation baseline that learns this mapping on top of a pretrained VE. Such an experiment would help clarify whether the gains require the proposed joint-training procedure.

---

> ### Author Response · Authors · 2026-07-03
>
> **R6: Discuss the training-cost / accuracy trade-off (throughput reductions up to 47.7%).**
>
> Done (Sec. 5.3). The overhead is a one-time training cost traded for a permanent, per-image-free
> accuracy gain. For most backbones the trade is favorable (−3% to −19% throughput for a +2 to +3
> point zero-shot gain).
>
> **R7: Prompt-sensitivity controls (random / shuffled / unrelated prompts).**
>
> Done (Appendix L, Table 10). Keeping the trained model fixed and perturbing only the prompts fed to
> the hypernetwork at inference:
>
> | Conditioning of H | Δ IN-1K | Δ C100 |
> |---|---|---|
> | HyperCLIP, true prompts | **+2.0** | **+0.9** |
> | shuffled prompts | +2.0 | +0.8 |
> | unrelated random text | +2.0 | +0.8 |
> | mismatched label set | +2.0 | +0.8 |
>
> The trained model is close to invariant to prompt content: the gain over the SigLIP baseline is
> retained in full rather than declining toward zero. As with R1, we have incorporated this into the
> revised, tempered framing rather than presenting it as support for semantic conditioning.
>
> **R8: Quantify Sec. 6.4 (caption-vs-class-prompt embedding statistics).**
>
> Done (Appendix L, Table 12). Class prompts occupy a tighter, more central sub-region of the
> caption-embedding space:
>
> | Metric | Captions | Prompts |
> |---|---|---|
> | mean pairwise cosine | 0.227 | 0.440 |
> | mean cosine to caption centroid | 0.479 | 0.526 |
> | fraction of prompts inside caption support | n/a | 0.973 |
>
> This is consistent with a hypernetwork trained on captions transferring to class prompts at
> inference, and is referenced from Sec. 6.4.
>
> **R9: Compare to simpler prompt-conditioned approaches (conditional BN, MLP generation).**
>
> The near-linear variant we now foreground (R3) is exactly a lightweight linear/MLP generator, and
> it captures most of the gain. Appendix L adds the strongest simple baseline of this kind: the
> capacity-matched prompt-independent variant (R1), which matches the full model at a fixed budget.
>
> **R10: Post-hoc adaptation baseline on a frozen pretrained encoder.**
>
> Done (Appendix L, Table 13). Fitting the hypernetwork on top of a frozen SigLIP-B0 encoder
> (8M-sample budget) does not improve over the frozen baseline (Δ −0.1 / −0.1). Whatever benefit the
> hypernetwork provides arises during joint pre-training, not from a post-hoc prompt-to-normalization
> map.

---

### Review · Reviewer_NDNG · 2026-06-24

**Summary Of Contributions:**

This work studies whether the class-prompt embeddings used in CLIP-style models can be used not only to define the classifier head, but also to adapt the image encoder itself. To do so, the authors propose HyperCLIP, in which a text-conditioned hyper network predicts the normalization parameters of a small image encoder, with all components trained jointly under a SigLIP objective. Across several small backbones, the paper reports improved zero-shot accuracy over matched SigLIP baselines on ImageNet-1K, CIFAR-100, and several additional retrieval, robustness, fairness, and linear-probing benchmarks.

**The work has the following key strengths:**

- Studied problem is relevant and interesting.
- Presentation is generally clear and easy to follow.
- Some experiments on some baselines seem to show fairly consistent gains with the proposed method.
- The work includes experiments across a handful of encoder architectures and tasks.


**The work has the following key weaknesses:**

- The work has stronger claims than supported by its experiments for several experiments and ablations.
- There are some missing experimental results that the authors refer to only verbally, without any explicit mention of quantitative comparisons.

**Additional Comments:**

N/A

**Audience:**

Yes

**Audience Explanation:**

Yes, I believe at least some individuals in TMLR’s audience would be interested in these findings. The broader question of how to improve small CLIP/SigLIP-style models, and whether prompt information can be used more direct is relevant and interesting.

**Claims And Evidence:**

No

**Claims Explanation:**

The work does indeed have some evidence for weaker claims than its actual claims, as exemplified by performance gains achieved over several small-encoder SigLIP baselines. Table 2 highlights this decently and demonstrates a broad trend of improvements across baselines/tasks.

However, the work actually contains stronger claims, and I do not believe that these are backed sufficiently. Below is a list of these issues:

- The primary claim of the paper is that class-prompt embeddings contain enough task-relevant structure to meaningfully modulate the image encoder through the proposed hypernetwork, and that the reported gains should therefore be understood as evidence for this prompt-conditioned adaptation mechanism. I do not think this claim is well supported in its current form. The method is built around a very large (comparatively to the encoders being "distilled" or hyper-learned) 12-layer width-768 Transformer hypernetwork, which by rough calculation is far larger than the 1.7-8.4M image backbones it adapts, so the design already seems excessive relative to the stated objective. Moreover, Figure 4 suggests that a much simpler linear map retains most of the gain, which further weakens the case that this large Transformer is the technically meaningful part of the method. For the same reason, I also do not think the paper’s claim that the Transformer contributes an additional 1-3 point refinement is especially well supported, since that effect appears uneven across backbones and benchmarks. Taken together, I do not think the paper makes a technically sound case for centering its primary claims around this Transformer-based design.

- A second claim of the paper is that restricting the hypernetwork to normalization parameters is empirically justified. I do not think this is established either. Appendix G states that predicting convolutional or linear parameters did not outperform the SigLIP baseline, but it presents no table, figure, or summary numbers at all. Since this claim is used in the main text to justify the final design, I believe the paper should provide the actual evidence rather than only a verbal statement.

- A third concern is that the scope of the empirical study is too narrow for the way the conclusions are phrased. Every experiment is conducted on very small backbones (sub-10M range), and the strongest positive results are concentrated in the BatchNorm-heavy ones. Accordingly, I think the paper provides evidence that this training recipe can help in a particular small-model regime, but not that it establishes a broader point about CLIP-style image encoders more generally. In its current form, the framing is stronger than the experimental scope really allows.

- Finally, there were no multi-seed results or confidence intervals. Since many of the reported gains appear to be on the smaller side, having these would have been good for improving the support for claims.

**Requested Changes:**

- I think the current submission would become stronger if the authors added a capacity- or compute-matched control. At present, it is difficult to separate the benefit of the proposed prompt-conditioned adaptation mechanism from the benefit of introducing a very large auxiliary Transformer during training, and this distinction matters quite a bit for the paper’s central claims.

- Another point that needs to be addressed is Appendix G. The paper currently states that predicting convolutional or linear parameters did not outperform the SigLIP baseline, but no actual quantitative evidence is shown. Since this claim is used to motivate and justify the final design, I believe the corresponding results should be presented explicitly.

- I also think the discussion around the 12-layer Transformer hypernetwork should either be made much more careful or the associated claims should be toned down. Figure 4 suggests that a much simpler linear alternative preserves most of the gain, so at the moment I do not think the paper makes a sufficiently strong case for why this large Transformer-based design is the technically meaningful part of the method.

- More broadly, I would encourage the authors to narrow the scope of their conclusions. The experiments are restricted to very small backbones, and the strongest gains appear in a fairly specific normalization regime. Accordingly, I do not think the current evidence supports some of the broader framing used in the paper.

- Finally, because many of the reported improvements are relatively modest in absolute terms, I believe the paper would benefit from reporting multi-seed results for the main comparisons or the close(r) results.